# Bacterial Carbonate Precipitation Using Active Metabolic Pathway to Repair Mortar Cracks

**DOI:** 10.3390/ma15196616

**Published:** 2022-09-23

**Authors:** Ali Raza, Rao Arsalan Khushnood

**Affiliations:** NUST Institute of Civil Engineering (NICE), School of Civil and Environmental Engineering (SCEE), National University of Sciences and Technology (NUST), H/12 Campus, Islamabad 44000, Pakistan

**Keywords:** permeability, bacteria, biodeposition treatment, mortar, image analysis, ultra-sonic investigation, recovered compressive strength, microstructure, RAMAN

## Abstract

A study was conducted to check the efficacy of microbial pathways for calcite precipitation to heal pre-existing cracks in mortar. In this experiment, realistic cracks of varying widths were induced on a mortar sample. Different repair methods were applied to a total of 22 mortar samples. Twelve cracked mortar samples with average crack widths ranging from 0.29 to 1.08 mm were subjected to biodeposition treatment using calcium lactate as a food source. The remaining ten samples were split into two groups: five cracked mortar samples were exclusively exposed to the bacterial solution, while the remaining five samples were immersed in distilled water. Digital image processing was used to extract the crack characteristics before and after the repair application. During several repair cycles, image processing was used to track healing. Further, these repaired mortar samples underwent examination for water permeability, ultra-sonic examination, and examination for recovered compressive strength. A forensic examination of the healing product in terms of morphology and elemental composition was conducted using RAMAN, XRD, SEM-EDS, and TGA. The water permeability of the repaired mortar sample using biodeposition with Ca-lactate was dramatically reduced, but samples treated with bacterial solution and distilled water did not exhibit any significant reduction. Complete crack healing was observed when using Ca-lactate as a food source for microbial repair. The forensic analysis confirmed the presence of calcite in healing precipitates

## 1. Introduction

The ultimate and most obvious problem in the life span of a concrete structure is cracking. Because concrete has low tensile strength and is brittle, it is prone to cracking. Furthermore, concrete is subject to cracking as a result of high strain (due to excessive or prolonged loading), creep, expensive reactions such as the Alkali aggregate reaction, thermal expansion, and harsh weather conditions [1]. Because hazardous solutions and gases can easily travel through these cracks, they impair the overall durability of concrete [2]. If not addressed immediately, these cracks can cause serious damage. Concerns regarding the preservation of existing concrete structures have prompted experts to look into developing repair technologies to slow or perhaps stop concrete degradation [3,4].

Self-healing is achieved via autogenous and autonomous healing. Autogenous healing is achieved due to anhydrous cement particles which cause secondary hydration via water infiltration through micro-cracks, whereas autonomous healing is achieved by using super absorbent polymer (SAP), admixtures, and microbes [5,6]. Microbially induced calcite precipitation (MICP) takes place using calcifying bacteria through metabolic activity [5]. Autonomous concrete crack-repairing techniques are further divided into two major categories. One method involves mixing bio-suspension into fresh concrete before cracks form, resulting in self-healing concrete [2,5,7]. Other concrete repair approaches include manually applying bio-suspension to existing cracks on concrete surfaces that lack a self-healing process [1]. Bio-mediated calcite precipitation (CaCO_3_) and salt bridge-building between particles are two benefits of microbial healing. After this healing mechanism, it is expected that the concrete’s integrity will be restored, and it will have appropriate resistance to water penetration.

The use of calcifying microorganisms in microbial precipitation has substantially enhanced the durability and mechanical strength of concrete [5]. The microbial pathway’s ability to repair the cementitious system is based on metabolic activity producing calcium carbonate, which is governed by nucleation sites, high concrete alkalinity, calcium ions, and dissolved inorganic chemicals. Furthermore, bacteria’s ability to precipitate bio-calcite is influenced by bacterial strain, pH, temperature, the precipitation process, and nutrient supply [5]. Calcium carbonate (CaCO_3_) precipitation is produced by bacteria that are either autotrophic or heterotrophic [8]. Heterotrophic bacterial processes are mostly studied because the bacteria can survive in the alkaline pH environment of concrete [9,10].

The use of ureolytic bacteria for calcification in self-healing concrete has been well documented [3]. In this passive pathway for calcification, the hydrolysis of the urease enzyme causes calcite (CaCO3) precipitation in these ureolytic bacteria. Ammonium (NH_4_^+^) and carbonate (CO_3_^−2^), as a result of the hydrolysis of urea, give rise to pH and carbonate concentrations (Equations (1)–(5)) [4,11]. The survival of these bacteria through direct intrusion is significantly hampered by the high alkalinity of the cementitious matrix, which severely limits the calcification capacity for self-healing [12]. Researchers have used a variety of immobilizers to protect microorganisms from the inhospitable environment, including silica gel, lightweight aggregate (LWA), and expanded clay [12,13]. Because of dense calcite (CaCO3) generation under negative zeta potential and the wide variety of bacteria that produce the urease enzyme, the urea hydrolysis process is widely used in self-healing concrete [11].

For instance, the ureolytic bacterial strain for self-healing *Bacillus sphaericus* was employed by P. Jongvivatsakul et al. [12] and J. Intarasoontron et al. [14], whereas S.S. Bang et al. [15] and H. Rong et al. [16] used *Bacillus pasteurii*. Ramakrishnan, who was also the first to bring microbial repair solutions to the concrete industry, used *Bacillus pasteurii* and *Sporosarcina* as self-healing materials to repair cracks [9]. The disadvantage of utilizing urea hydrolysis is that the by-product, ammonia, is damaging to the environment and the cementitious matrix [17]. Each carbonate ion produces two ammonium ions (NH_4_^+^), polluting the environment with nitrogen. Furthermore, both the urease enzyme and urea cannot last for years, reducing their healing capacity [18]. Ureolytic bacteria’s metabolic conversion is effective at pH 9, which is significantly hampered by the high alkaline habitat of concrete [19]. Furthermore, an immobilizer was used to protect microorganisms from high-alkaline concrete settings. The need for an immobilizer/carrier complicates the repair process, increases the overall cost of repair, and the immobilizer may react and perhaps kill microorganisms [10].
(1)$$\mathrm{CO}{\left({\mathrm{NH}}_{2\;}\right)}_{2}+H_{2}O\to {\mathrm{CO}}_{2}+2{\mathrm{NH}}_{3}$$
(2)$${\mathrm{NH}}_{3}+H_{2}O↔{\mathrm{NH}}_{4}^{+}+{\mathrm{OH}}^{-}$$
(3)$${\mathrm{CO}}_{2}+{\;\mathrm{OH}}^{-}\to {\;\mathrm{HCO}}_{3}^{-}$$
(4)$${\mathrm{HCO}}_{3}^{-}+{\mathrm{OH}}^{-}\to {\;\mathrm{CO}}_{3}^{2-}+H_{2}O$$
(5)$$\mathrm{CO}{\left({\mathrm{NH}}_{2\;}\right)}_{2}+2H_{2}O\to {\mathrm{CO}}_{3}^{2-}+2{\mathrm{NH}}_{4}^{+}$$

Another process for bio-mediated self-healing is called the active metabolic pathway; this involves the oxidation of organic acid, which raises pH and dissolves inorganic carbon. The release of CO_2_ due to the aerobic oxidation of organic acid encourages the synthesis of carbonate (CO_3_^−2^), which combines with abundant calcium present as a food source to produce bio-mediated calcite (CaCO_3_) [2,5,17]. Because ammonia and sulfide are not present, this self-healing approach is more sustainable. Many researchers have used an active metabolic pathway for calcite precipitation; H.M. Jonkers et al. [20], N. Shaheen et al. [10], and M. Kanwal et al. [2] used *Bacillus subtilis*, while V. Wiktor used *Bacillus alkalinitrilicus* for self-healing concrete. These *Bacillus* species may remain as dormant cells, but when cracks emerge in the concrete’s surface, the bacteria’s spores become activated when they come into contact with penetrating oxygen and water [7]. Furthermore, alkaliphilic bacterial species have a high level of alkaline resistance. At pH 10, these microbial species have higher bio-mediated calcite (CaCO_3_) potential, indicating their alkaliphilic nature and compatibility with the concrete environment [5]. In the active bio-mediated calcification process, the presence of concrete-compatible feed and microbial solution yielded six molecules of calcium carbonate without any dangerous by-products (Equations (6) and (7)). According to research, the increase in pH during bacterial carbonate precipitation favors the conversion of atmospheric CO_2_ into calcite (CaCO_3_). As a result, bio-mediated calcification for self-healing could be a viable alternative for CO_2_ removal from the atmosphere [21]. This method for self-healing is relatively safe and eco-friendly. Microbial precipitation for self-healing using these methods (active and passive) bonds particles together and fills up cracks and fine pores. Microbial precipitation provides bonding to loose materials via bio-mediated calcite (CaCO_3_) and salt bridge formation between particles [22].
(6)$$\mathrm{Ca}{\left(C_{3}H_{5}O_{3}\right)}_{2}+6O_{2}\to {\mathrm{CaCO}}_{3}+5{\mathrm{CO}}_{2}+5H_{2}O$$
(7)$$5{\mathrm{CO}}_{2}+\mathrm{Ca}{\left({\mathrm{OH}}_{\;}\right)}_{2}\to 5{\mathrm{CaCO}}_{3}+5H_{2}O$$

In recent years, however, tremendous progress has been made in the field of self-healing concrete. All of the studies, however, are focused on living concrete made by combining the microbial solution with traditional concrete materials during the mixing process. The efficiency of these approaches in the repair of pre-existing cracks on non-living concrete surfaces with no bio-self-healing mechanism is rarely investigated. The available studies are either very limited or rely on passive urea hydrolysis mechanisms. The strong urease activity of *Bacillus pasteurii* with urea-CaCl_2_ feed was used by S.-G. Choi et al. [22] for crack repair up to 1.64 mm. In conjunction with complete crack closure, enhanced watertightness was also achieved via the external application of these ureolytic bacterial repair solutions. *Bacillus sphaericus* immobilized with silica gel was used by K. Van Tittelboom et al. [1] for crack repair up to 0.87 mm. Based on improved permeability and crack sealing, the epoxy-based repair was compared to ureolytic bacterial repair. *Bacillus sphaericus* was also used by W. De Muynck et al. [3] for surface coating, leading to enhanced endurance against water permeation. An artificial crack of width 0.3 mm, a 20 mm depth, and a 50 mm length were repaired by Abo-El-Enein et al. [23] using *Sporosarcina pasteurii* with urea-CaCl_2_ feed. Complete crack remediation with enhanced durability regarding rapid chloride permeability and improved mechanical performance with regard to compressive strength was achieved. As previously stated, urea hydrolysis releases ammonia, which is harmful to the environment and the cementitious matrix. Additionally, both the urease enzyme and urea are short-lived, limiting their ability to repair [17,18,24]. The metabolic conversion of ureolytic bacteria is optimal at pH 9, which is substantially impeded by the high-alkaline environment of concrete. An immobilizer was also used to protect microorganisms from alkaline concrete environments. The use of an immobilizer/carrier complicates the repair process, raises repair costs, and the immobilizer may react with bacteria, possibly killing them.

Based on the findings of the above studies, a more robust microbial healing technique is needed which is both sustainable and durable in a high-alkaline concrete environment without the use of an immobilizer [25,26]. The purpose of this research is to see microbial repair efficiency through an active metabolic pathway. The *Bacillus pumilus* strain isolated from soil was chosen for this study because of its excellent performance in a highly alkaline environment and dense calcite production [5]. Due to its efficacy in repair and high compatibility in self-healing concrete, calcium lactate was employed as an organic food source for calcite precipitation [17]. For the repair of mortar cubes, realistic cracks were created in this experiment. Realistic cracks have a crack width of 0.29–1.29 mm on average. Water permeability, compressive strength recovery, and ultra-sonic investigation were used to assess the effectiveness of the microbial crack repair. As a defense mechanism, bio-mediated calcite (CaCO_3_) provides great durability against water penetration, increased mechanical strength, and microstructural refinement. Several microstructural tests, such as SEM-EDX, RAMAN spectroscopy, and TGA analysis, were used to confirm the presence of the bio-mediate calcite.

## 2. Materials and Methods

### 2.1. Microbial Solution

The selected Gene of *Bacillus* species for this study was *Bacillus pumilus* (accession No. #MN865840, NCBI GenBank), which has negative urease activity [5]. *Bacillus pumilus*, typically found in soil, is a Gram-positive, rod-shaped, and endospore-forming bacterial strain. Because of the production of protective endospores, it can withstand high pH, making it excellent for cementitious repair. The stocks of strain were revived from a −80 °C glycerol solution and streaked on nutrient agar plates with pH 7.5, comprised of 5 g peptone, 15 g agar, 2 g yeast extract, 5 g chloride, and 1 g Lab-Lemco Powder. The plates were stored for 24 h in a static incubator at 37 °C. Microbial colonies emerged on Petri dishes after 24 h. For the germination of bacterial culture, Tryptone soy broth (TSB) medium with pH 7.3 ± 0.2 at 25 °C was autoclaved at 121 °C for 15 min. The inoculum of the bacterial colony from the Petri dish was inoculated in TSB medium, comprised of 3 g papaic digest of soybean meal, 2.5 g dextrose/glucose, 17 g pancreatic digest of casein, and 2.5 g dipotassium hydrogen phosphate. Then, the cultured medium was moved to a shaking incubator at 37 °C. To set the optimum growth concentration, the growth characteristics of *Bacillus pumilus* in the TSB medum were tabulated, using the standard protocol by Prescott et al., 1999 [27]. The detailed microbial growth curve preparation is presented in Figure 1, and started with growing the microbial culture on nutrient agar plates. After that, the fresh bacterial culture was transferred to the broth solution. The solution was transferred to the shaking incubator set at 120 rpm and 37 °C, followed by subsequent measurement of the optical density value. Along with OD measurement the cell concentration value corresponding to each OD value was also measured using the serial dilution method. Those OD and cell concentration values are plotted in Figure 2.

After capturing the growth characteristics of *Bacillus pumilus*, a microbial repair solution was devised by inoculating a freshly prepared bacterial colony in autoclaved TSB medium and transferring it to the shaking incubator for 18–20 h at 37 °C, at 120 rpm. The optical density of cultured medium was measured using a SPECORD 200 plus spectrometer at a 600 nm wavelength. The microbial repair solution used in this study had an OD_600_ value of 1.3, with a bacterial concentration of 2.9 × 10^10^ Cells/mL. The complete protocol followed for bacterial repair preparation is presented in Figure 3. Previous investigations and bacterial strain growth characteristics were used to determine the OD value and cell concentration [5,27,28].

### 2.2. Calcium Lactate ${Ca(C}_{3}H_{5}O_{3}{)}_{2}$ Solution

In an active metabolic pathway for bio-mediated calcite precipitation, a continuous supply of a calcium-rich environment is necessary. According to Jonkers et al. [20], Shaheen et al. [10], and M. Kanwal et al. [2], among many Ca-based sources, calcium lactate was found to be the most compatible food source. Further previous research also suggests that the combination of the *Bacillus* species with calcium lactate resulted in dense calcite formation [2,5,7]. For this study, organic food was comprised of the calcium lactate solution ${\mathrm{Ca}(C}_{3}H_{5}O_{3}{)}_{2}.$ A total of 80 g of Ca-lactate was mixed with 1000 mL of distilled water to make an organic food solution. After mixing, the solution was autoclaved for 15 min at 121 °C.

### 2.3. Mortar Samples

Siliceous sand, ordinary Portland cement CEM-1 (Grade 53), and distilled water were used to make mortar samples. OPC has a specific gravity of 3.14 and a mean particle (D50) size of 9.5 µm, confirming ASTM C-150 [29]. Table 1 and Table 2 list the chemical and physical properties of CEM-1. X-ray fluorescence was used to determine the chemical makeup of OPC. The fine aggregate was siliceous sand with a fineness modulus of 2.6 and a specific gravity of 2.64, sourced from a local source (Table 3). The water-to-cement (*w*/*c*) ratio was 0.485, while the sand-to-cement ratio was 2.75. The flow value obtained after 25 drops on the flow table was 95–100 in. By mixing all these ingredients, a mortar specimen of 50 × 50 × 50 mm was made under ASTM C-109 as shown in Figure 4 [30]. The molds were placed in a controlled environmental chamber for 24 h after casting, with an air-conditioned temperature of 20 °C and relative humidity of 90%. The samples were demolded and immersed in a water storage tank on completion 24 h after casting. The reference sample’s compressive strength was measured at 42.616 N/mm^2^ at the end of the 28th day, with a standard variation of 3.2 N/mm^2^.

### 2.4. Crack Formation and Image Analysis

On the 28th day after casting, the mortar specimens were withdrawn from the water tank to obtain distinct crack features. The specimens were left on the table for 10–15 min to allow the surface moisture to evaporate. The samples were then placed in a compression machine, with a steel nail positioned laterally in the center of the top surface. Figure 5 shows a schematic of fracture formation in a mortar cube. When an obvious surface crack appeared, loading was halted. After the cracks had formed, the samples were ready for repair.

Healed crack areas represent the healing impact of the repair solution. It is critical to assess the healed area to determine the effectiveness of the repair process. A digital image processing approach was used to extract the crack features [31,32,33,34,35]. Before the application of any repair, high-definition images were captured with subsequent crack feature extraction. After loading the image into MATLAB, an image processing algorithm was applied. Initially, an RGB (Red–Green–Blue) image was transformed into a binary image using OTSU-based image thresholding algorithms as shown in Figure 6 [33,36]. The image was cleaned after the crack region was identified. No-crack objects, noisy pixels, and places other than the intended crack were deleted, as well as the picture’s edge smoothing. After removing unwanted pixels, the crack width (mm) and crack area percentages (%) were calculated using Equations (8) and (9). Because measurements on the edges are questionable, the core 40 mm crack length was evaluated, and a 5 mm portion on the edges was removed from the parameter acquired via boundary extraction. The complete protocol for image processing is shown in Appendix A.
Average Crack Width (mm) = (ACA_avg_ (mm^2^))/(ACL_avg_ (mm))ACA_avg_ (mm^2^) = Average Crack Area.ACL_avg_ (mm) = Average Crack Length.(8)
Crack Area (%) = (CA (mm^2^)/SCA (mm^2^)) × 100%CA (mm^2^) = Crack Area.SCA (mm^2^) = Sample Cross-Section Area.(9)

### 2.5. Microbial Crack Repair

After digital image processing was used to seize the crack feature on the mortar surface, a bio-consortia for microbially induced calcite precipitation (MICP) was used. In this study, two different biodeposition suspensions were made to check the repair efficacy of microbial repair. The germination of the bacterial solution and the preparation of the food source solution were discussed in Section 2 and Section 2.2, respectively.

In the first type of repair solution, bacterial repair comprised only the bacterial solution in the absence of any organic food source (Appendix A). The procedure observed in that repair was as follows:The mortar crack sample was submerged in 100 mL bacterium solution in a sterilized plastic cup for 4 h.The plastic cup was covered using cling film to avoid any external contamination.On completion after 4 h, the samples were removed from the plastic cup and left to dry for 24 h.The protocol was repeated from step (1) to step (3) for the next cycle.

In the second type of repair solution, bacterial repair treatment is comprised of both the bacterial solution and the food source (Appendix A). The protocol observed in that repair was as follows:The cracked mortar samples were immersed in 100 mL bacterium solution in a sterilized plastic cup for 4 h. The plastic cup was covered with cling filmOn completion after 4 h, the samples were placed on the table for 10 min and the surface was wiped using a paper towel.All the repair samples were soaked in the container of Ca-lactate solution for 24 h.The samples were removed from the food source and left to dry on the table for 15 min.The protocol was repeated from step (1) to step (4) for the next cycle.

All biodeposition repairs were carried out in a temperature-controlled-environment chamber at 25 °C. The above-mentioned procedure was defined as a single MICP repair cycle.

### 2.6. Crack Repair Investigation

To assess the efficacy of the repair solutions, the water permeation ability of the cracked mortar samples was assessed on completion of 0, 5, and 7 cycles of the “Bac. pum. No Ca”, and on completion of 0, 7, and 14 cycles of the “Bac. pum. Ca” formulations of bio-deposition. The image analysis technique was used to evaluate percentage repair after the completion of different cycles of bio-deposition. Ultra-sonic values and the recovered compressive strength were also assessed for all the crack repairs (Table 4). Microstructural analysis of the healing material extracted from the crack surface as the result of bio-mediated precipitation was also performed using SEM-EDX, RAMAN, XRD, and TGA techniques.

### 2.7. Water Permeability

A continuous water head permeability test was performed on the mortar samples to measure the efficiency of the various repair treatments. This is a modified version of S.-G. Choi et al. [22] and K. Van Tittelboom et al. [1] water permeability test setup. The permeability was measured after each cycle of different repair formulations. The repair cycles of bio-deposition were discussed in Section 2.5. For samples treated with water, the samples were treated by immersing them in a distilled water tank for 4 h. The samples were dried in the open air for 24 h. This was considered to be one water treatment cycle. The permeability was evaluated after the completion of 0, 7, and 14 cycles of treatment with water. The schematics of the permeability apparatus are given in Figure 7.

Initially, the samples’ crack features were assessed using image processing. The samples were then submerged in water for 24 h before being tested for permeability. The test specimen was wrapped with ultra-sealing tape for to render it watertight after soaking, leaving the top and bottom surfaces exposed. The plexiglass container was fixed to the top of the specimen after sealing, as shown in Figure 7. Watertightness was ensured by placing a rubber seal between the plexiglass and the sealed sample. A water tightening seal was wrapped around the sample and plexiglass after it was fitted to make the total assembly completely waterproof. The plexiglass-mounted sample was put on a supporting platform inside a water-filled tank. By mounting a hole for water flow in the upper compartment and a water container for penetrated water, a steady water head was maintained. A constant supply of water was maintained throughout the experiment. The amount of water flowing out of the sample was measured at 2 h, 4 h, and 6 h intervals to estimate permeability using Equation (10):(10)Kms=QA×T×ΔL/L
where, Q represents the total volume of penetrated water collected, the cross-section area of the sample is represented by A, the constant water head is represented by ∆L, T is the time taken, and L is the highest value of the sample. All the calculations were performed in metric units.

### 2.8. Healing and Watertightness Evaluation

Crack sealing and enhanced watertightness parameters were used to check the effectiveness of biodeposition repairs. Before conducting crack restoration, the crack features were analyzed utilizing digital image processing for crack sealing evaluation (indicated in Section 2.4). After the application of biodeposition repair cycles, the repaired crack features were examined using Equation (11). *A*_0_ is the crack area before repair and *A*_1_ is repaired crack area.

The improved watertightness after the repair was evaluated using the permeability values of the samples. Before any repairs, the samples’ water permeability was assessed. Watertightness was determined by permeability measurements after the repair cycles of biodeposition treatment using Equation (12). P_0_ is the initial permeability and P_1_ is the permeability value after repair.
(11)$$CSP\;\left(\%\right)=\frac{A_{0}-A_{1}}{A_{1}}\times 100\%$$
(12)$$\mathrm{WT}\left(\%\right)=\frac{P_{0}-P_{1}}{P_{1}}\times 100\%$$

### 2.9. Ultra-Sonic Pulse Velocity (UPV) Evaluation

This test was utilized in this investigation to see how different repair cycles affected the speed of wave propagation. In comparison to the cracked sample, waves can travel much quicker (4000–4600 m/s) in pristine (un-cracked) mortar specimens [1,12,37]. Each crack repair cycle of biodeposition results in crack sealing, which ensures that waves flow via bio-mediated precipitate rather than open fissures. To assure transmission between two ends, a pair of transducers is required. The sonic values were determined by placing transducers perpendicular to the cracked mortar surface under standard specification by ASTM C-597 [38]. This ultra-sonic experiment yielded the pulse arrival time, which is the amount of time it takes for a pulse to travel from one end to the other. An average triplicate ultra-sonic measurement was taken after each repair cycle. Controls’ Model 58-E4900 PULSONIC with a 54 kHz operational frequency was utilized for ultra-sonic evaluation.

### 2.10. Recovered Compressive Strength

At 28 days, the compressive strength of completely intact mortar samples was determined under ASTM C190 [30]. For repaired mortar samples, this test was used to check the percentage that recovered in compressive strength following the 28th-day strength. On completion of the repair cycles, the repaired mortar samples were dried for 24 h in a controlled-environment chamber at a temperature of 25 ± 2 °C and relative humidity of 50–60%. After that, the compressive strength was determined using ASTM C109 and the recovered compressive strength was obtained using Equation (13). *C*_28th_ is the compressive strength of the uncracked mortar sample on the 28th day and *C_re_* is the compressive strength after repair.
(13)$$CSR\;\left(\%\right)=1-\left(\frac{C_{28\mathrm{th}}-C_{re}}{C_{re}}\right)\times 100$$

### 2.11. Microstructural Confirmation of Calcite

As bio-mediated precipitate acts as the front line for durability enhancement of the repaired mortar samples, the resulting biodeposition via microbial activity was subjected to RAMAN spectroscopy, X-ray diffraction, SEM-EDX, and thermogravimetric analysis.

For RAMAN spectroscopy, the mineral composition and chemical structure of the biodeposition material were recorded using a BWS415-532S RAMAN spectroscope with a CleanLaze excitation laser at 532 nm. The spectra were noted in the spectral range of 94–1800 cm^−1^ using a 100× microscope with an excitation wavelength of 532 nm. Further, the obtained spectra were compared with the RRUFF database for pure calcite, and the captured modes were also compared with the published literature [39,40,41,42].

For XRD, a Bruker D8 was used to recode the XRD-spectra of the precipitate. Due to possible high thermal activity during the acquisition of XRD-spectra, the X-ray target was made of Copper (Cu) due to its ability to cool rapidly and produce strong Kα and Kβ lines [10]. The spectra were recorded from a 3 to 60° orientation angle (2θ) with a wavelength of 1.54 Å. The X’Pert HighScore software suite was used to analyze the outcomes of the analysis. In addition, the published literature and the Joint Committee of Powder Diffraction Standards (JCPDS), card number 5-586, supported the diagnosis of crystalline products [5,43].

For SEM-EDX, an FEI Nova 450 NanoSEM was used to obtain SEM micrographs of the biodeposition material. Sputter-coating was conducted using gold, with subsequent capturing images at 10 µm, 5 µm, and 1 µm. Further, spot-EDX was used to check the elemental mapping and chemical composition of crystals under SEM-micrographs. The results of the SEM-micrographs and EDX elemental mapping were further compared with the published literature [5,10,28].

For thermogravimetric analysis (TGA), weight loss occurred with increasing temperature. TGA is a type of thermal analysis that looks at how a substance’s chemical and physical properties vary as temperature varies. The calcite crystal disintegrated when heated between 600 and 850 °C, resulting in weight loss due to CO_2_ release (Equation (14)) [1]. The biodeposition material was heated between 0 and 1000 °C at a rate of 10 °C/min and resulted in mass depletion compared to earlier studies [1,2].
(14)$${\mathrm{CaCO}}_{3}\left(s\right)\to \mathrm{CaO}\left(s\right)+{\mathrm{CO}}_{2}\left(g\right)$$

## 3. Results and Discussion

Appendix A show the water permeability, watertightness, ultra-sonic measurement, healed crack area (%), and recovered compressive strength results on cracked mortar samples repaired using biodeposition and distilled water treatment. The sample IDs for the individual formulations were arranged based on average crack width from smallest to largest. Appendix A shows the top surfaces of the mortar samples treated with biodeposition repair.

Figure 8 represents a strong linear relationship between crack width (mm) and crack area (%), obtained from cracked mortar samples to be treated with biodeposition and distilled water treatment using image processing. This, consequently, leads to the use of average crack width as a representative crack size for the data analysis of all the formulations under study. The maximum and minimum crack sizes for biodeposition repair were 0.29 mm and 1.08 mm, respectively, while the maximum and minimum crack sizes for the water-treated formulation were 0.46 mm and 0.93 mm.

### 3.1. Healing Assessment

All of the cracked sample’s top surfaces were repaired utilizing biodeposition and distilled water channels, and the extent of healing is shown in Appendix A. Image processing and an optical microscope were used to assess the healed crack area after various cycles of restoration procedures.

#### 3.1.1. Digital Image Processing

As shown in Appendix A, the crack area features were evaluated for mortar samples treated with biodeposition repair with a Ca-based food source. The majority of the crack regions were repaired following seven cycles of biodeposition repair, as shown in Appendix A. In Appendix A, a binary picture obtained utilizing image processing depicts the aerial characteristics of sealed and unsealed regions of the cracks. All of the cracked mortar samples were healed after 14 cycles (Appendix A). Calcite precipitation, as shown in Appendix A, is responsible for the complete closure of cracks on mortar surfaces after the repair. This precipitation is primarily caused by the oxidation of organic acid (Ca-lactate) and the generation of carbonate (CO_3_^−2^), which eventually leads to calcite precipitation due to the presence of calcium (Equation (6)). In addition, existing Ca(OH)_2_ interacted with CO_2_ and transformed it into calcite (Equation (7)).

However, for samples treated without a Ca-based food source, the extent of healing after five and seven cycles was very low due to the lack of a Ca-based food source in biodeposition repair (Appendix A). This is mostly due to the lack of an excess Ca source available for calcite precipitation by bacteria. The maximum percentage of the healed area was only 11%, which is extremely low. As previously stated, the microbial metabolism releases CO_2_, which increases carbonate synthesis (CO_3_^−2^) [44]. Calcite is formed when these carbonates react with calcium. The lack of a Ca source in this formulation did not satisfy the constraint. Autogenic healing—which can occur as a result of the hydration of unhydrated cement particles inside the cementitious matrix, soluble hydrate deposition, or Ca(OH)_2_ carbonation—could be the cause of the small permeability reduction [1]. The minor permeability drop could be because the samples were just one month old before treatment. The SEM image in Figure 9 reveals the presence of Ca(OH)_2_ and unhydrated cement particles. In the presence of Ca(OH)_2_, CO_2_ is released via microbial metabolism and aid carbonation.

Crack healing was exceedingly slow in samples treated with distilled water. The cracks on the samples’ surfaces were visible even after seven cycles, as seen in Appendix A. Autogenic healing due to the hydration of anhydrous cement particles accounts for up to 4% of minor crack healing. Autogenic healing was ineffective for full sealing because the minimum crack width recorded in the samples was 0.46 mm. According to the literature, autogenic healing is quite successful in fixing cracks up to 0.2 mm in length [22].

#### 3.1.2. Optical Microscope

An optical microscope with a minimum count of 0.2 µm was utilized to evaluate crack width and track healing over different repair cycles. The crack width value at different locations, measured using optical microscopy, is mentioned in Appendix A. The measured crack width values were also compared to the crack width value determined via image processing (Table 5). In addition, Figure 10 shows the monitored healed fracture areas after several cycles of biodeposition and distilled water repairs. Progressive autonomous healing and an increase in calcite precipitates were seen with each increasing cycle of repair, resulting in fracture sealing. This self-healing was particularly noticeable in samples treated with biodeposition in the presence of a Ca-based food supply.

### 3.2. Permeability Assessment

After careful assessment of the permeability of mortar samples, the permeability was reduced after the application of different repair treatments, but the level of reduction varied depending upon the type and healing efficacy of repair. Figure 11 illustrates the permeability values of mortar samples using different repairs, with all the permeability coefficient values plotted on a logarithmic scale. From Figure 11, it is clear that permeability values increase with increasing crack width on the mortar surface. This increase in permeability values was more prominent for samples before biodeposition treatment.

For samples to be treated with biodeposition repair with a Ca source, the permeability values vary from 9.460 × 10^−7^ m/s to 1.822 × 10^−4^ m/s as the average crack width of mortar samples increases from 0.29 mm to 1.08 mm (Appendix A). The increasing trend of permeability with increasing crack width before the microbial repair is consistent with published literature by S.-G. Choi et al. [22] and K. Van Tittelboom et al. [1]. S.-G. Choi et al. [22] reported increasing permeability values from 9.24 × 10^−6^ m/s to 3.03 × 10^−3^ m/s for mortar cylinders with crack widths from 0.15 to 1.64 mm. In this investigation, more cracked areas were repaired after 7 and 14 cycles of biodeposition with a Ca source, resulting in a permeability range of 1.253 × 10^−4^~9.299 × 10^−7^ m/s after 7 cycles, which was further lowered to 7.439 × 10^−7^~9.359 × 10^−6^ m/s after 14 cycles (Appendix A). The permeability reduction after 14 cycles was three orders lower than the permeability of the untreated sample (0 cycles), with the first 7 cycles showing the most significant drop. Calcite precipitation resulted in the complete closure of fractures on mortar surfaces, in addition to a reduction in permeability (Appendix A). This precipitation is mostly due to the oxidation of organic acid and the generation of carbonate, ultimately leading to calcite precipitation due to a calcium-rich environment, as enumerated in Section 3.1 of the healing assessment. Furthermore, existing Ca(OH)_2_ interacted with CO_2_ and transformed it to calcite. As a result, the creation of a calcite layer on the cracked sample’s surface obstructed water entry, resulting in a fall in the permeability measurement, as illustrated in Figure 11a. The lowest permeability coefficient achieved after applying a microbial solution with a Ca source was 7.439 × 10^−7^ m/s, which is significantly greater than the permeability of virgin mortar, which is 1.0 × 10^−11^ m/s [45].

For samples to be treated with biodeposition without a Ca source, the permeability values varied from 1.031 × 10^−5^ m/s to 1.0182 × 10^−4^ m/s as the average crack width of mortar samples increased from 0.69 mm to 1.03 mm Appendix A). In said formulation, the decrease in water permeability after seven cycles was very low. This is primarily due to the absence of a food-source or an excess Ca source for calcite precipitation. In active metabolic pathways for calcite precipitation, the absence of a calcite source hampered the production of calcite [5]. The small reduction in permeability could be due to autogenic healing due to the hydration of unhydrated cement particles. The possibility of autogenic healing was already confirmed using SEM micrographs (Figure 9) in earlier discussions (Section 3.1). Similarly, for samples to be treated with distilled water, the permeability values varied in the range of 9.05 × 10^−6^~1.51 × 10^−5^ m/s of mortar samples, with an average crack width increase from 0.46 mm to 0.93 mm (Appendix A). The permeability reduction of samples treated with distilled water was not significant. The small reduction is attributed to autogenic healing.

The slope of the permeability coefficient and the average crack width of the mortar samples for biodeposition repair using a Ca source were lower than the reference samples, as shown in Figure 11a. When compared to the other repairs, the reduction in the slope was substantially more significant for this formulation. This means that bio-mediated calcite precipitation, rather than average crack width, controls the reduction in the permeability of the repaired samples. The R^2^ value of the fitted regression curve for the untreated sample is significantly higher, as shown in Figure 11a. The R^2^ value lowers with each biodeposition treatment cycle. Calcite precipitation is thought to be responsible for the decrease in R^2^. The amount of bio-mediated calcite and its microstructure determine the permeability reduction after a particular period of treatment with biodeposition. The network formation of calcite precipitate inside the crack surface resulted in a considerable decrease in R^2^ values for the mentioned formulation. The decrease in R^2^ is consistent with previous research [22]. In biodeposition repair comprised of the only bacterial solution, the smaller reduction in the slope in Figure 11b suggested less calcite precipitation.

Equation (12) was used to assess watertightness using permeability values from various treatments (Appendix A). The improved watertightness of the specimen was ensured by the lowered permeability value after each repair session. Appendix A shows the watertightness of samples treated with various repair solutions. The majority of the specimens for biodeposition repair using a Ca source demonstrated significant watertightness. The drop in permeability for samples with high crack regions (%) is substantially greater than for samples with small crack areas (%) [22]. As a result, crack widths greater than 0.7 mm resulted in a considerable increase in watertightness for the sample treated with “Bac. Pum. Ca” repair. With each successive healing cycle, the rate of permeability reduction decreased noticeably. Thus, seven cycles of biodeposition repair with Ca source were used to attain optimum watertightness. Because there was less permeability reduction in the sample treated without a Ca source, the watertightness recovery was also low. Similarly, there was no substantial recovery of watertightness in samples treated with distilled water.

A comparison of recovered watertightness via biodeposition repair with and without a Ca source is shown in Figure 12. Because of the wide range of crack sizes, the samples were separated into groups depending on the crack area (%). Biodeposition-treated specimens yielded samples with crack area percentages of 1–1.5 % and 1.5–2.5 %. Figure 12 shows the water-tightness of samples with crack area percentages ranging from 1 to 1.5 %. It can be noticed that samples treated with biodeposition repair with a Ca source in seven cycles, and samples treated without a Ca source in seven cycles, exhibited watertightness of up to 60% and 5%, respectively. Whereas samples with a crack area percentage of 1.5–2.5%, as shown in Figure 12, showed 20–90% and 5–20% watertightness, respectively. Measured permeability is directly connected to the healed area, and watertightness is a function of measured permeability. As a result, for biodeposition repair using a Ca source, improved watertightness was achieved due to complete crack closure.

### 3.3. Ultra-Sonic Assessment

The ultra-sonic investigation was conducted before and after the application of biodeposition and distilled water repair treatments. The effectiveness of each repair was evaluated using this test by comparing results. Firstly, the sonic value was calculated before inducing cracks on the mortar samples. As shown in Appendix A the ultra-sonic value for pristine samples varied between 3600 and 4450 m/s, implying that the samples were of good quality [12,37]. The trends of ultra-sonic values after the application of different repair cycles are also plotted in Figure 13.

Before the repair treatment of biodeposition with Ca source, the ultra-sonic values for cracked samples were lowered by up to 2116.67–3078.79 m/s, a 50–70% drop in ultra-sonic value compared to the control uncracked samples (Appendix A). The existence of a crack on the mortar surface causes an increase in sonic wave arrival time, resulting in a drop in ultra-sonic value [1]. After the application of said repair, the sonic value was increased to 2902.86–3907.69 m/s, showing a recovered ultra-sonic value of up to 60–88% of the uncracked sample (Figure 13). Comparing the results for samples treated with biodeposition repair with a Ca-based solution led to the conclusion that a significant reduction in sonic wave transmission time was recorded. Calcite precipitation by alkaliphilic bacteria causes crack closure in samples treated with bio-consortia with a Ca-based food source. The entire cementitious matrix becomes denser as a result of this calcification, lowering the elapsed period of the sonic pulse. The increase in ultra-sonic measurement as the number of cycles of microbial treatments increases is consistent with previous research [1].

Changes in ultra-sonic value for samples treated with biodeposition without Ca-source repair are displayed in Figure 13. Before treatment, the ultra-sonic values of the sample with said repair ranged from 3907 to 4198 m/s, indicating a reduction of nearly 50–55% compared to the control sample (Appendix A). After the treatment, the obtained ultra-sonic values ranged from 2000 to 2300 m/s, indicating that the recovered ultra-sonic value was only 50–60% of the control sample. The mortar’s overall quality did not improve as much as it did with previous biodeposition treatments. Primarily this is because of the absence of a Ca-based food source, resulting in less calcification and crack closure. Similarly, the ultra-sonic measurement for distilled water-treated samples is shown in Figure 13. The recovered ultra-sonic value was only 52–55% of the control sample, which is relatively low because this treatment mostly involved autogenic healing (Appendix A).

### 3.4. Recovered Compressive Strength

Compression strength is the basic desired property of a cementitious matrix. Upon completion of the biodeposition and distilled water repair treatments, samples were subjected to a compressive strength test under ASTM C109 [30]. The measured compressive strength value of the pristine mortar sample on the 28th day was 42.616 N/mm^2^. On the contrary, the compressive strength of the repaired cracked mortar samples was low, but the extent of recovery varied depending upon the type of repair (Appendix A).

The maximum compressive strength was recovered from samples treated with biodeposition repair with a Ca source, as indicated in Appendix A. Other formulations, such as those using a bacterial solution only and distilled water, yielded lower compressive strength recoveries. As shown in Figure 14, compressive strength was regained using “Bac. Pum. Ca” repair in the range of 61–79%. As previously stated, each increasing cycle increases the amount of calcite precipitation on the sample surface and within the cementitious matrix, resulting in greater compactness [2]. The sample’s compressive strength increases as the compactness increases, whereas the recovered compressive strength was 40–55% for the sample treated with distilled water and 47–56% for the sample treated with BS repair, respectively (Figure 14).

### 3.5. Microstructural Confirmation of Calcite Formation

Carefully extracted healing product from the cracked mortar surface after biodeposition treatment was subjected to RAMAN spectroscopy, XRD analysis, SEM-EDX, and TGA to confirm the calcite precipitation forensically. RAMAN spectroscopy is a non-destructive technique for detecting vibrational modes and determining the structural fingerprints of molecules [46]. The healing material collected from the crack location was subjected to RAMAN analysis, which revealed distinctive calcite peaks (Figure 15). The singlet signal at 1087 cm^−1^ was produced from the internal-mode A_g_, derived from v_1_ stretching of the carbonate ion of the healing material [47]. Other low-intensity peaks are found below 300 cm^−1^ and 710 cm^−1^, respectively, due to translational and in-plane bending modes [48]. According to the literature, the peaks found in this study using RAMAN analysis were distinct from other polymorphs of calcite, such as vaterite and aragonite, demonstrating that calcite is the only stable form of calcium carbonate [46,48]. Pure calcite spectra acquired from the RUFF were used to confirm these results (#R040070) [42].

The XRD of the healing precipitates was recorded from a 3 to 60° orientation angle (2θ) with a wavelength of 1.54 Å. The obtained peaks from XRD analysis were used to study the morphology and phase purity of the collected healing product. Furthermore, a comparison was drawn with a sample that was not treated with biodeposition treatment. The obtained peaks from the untreated samples were primarily composed of quartz, ettringite, portlandite, low calcite, etc as given in Figure 16. The presence of low calcite content in the untreated sample, with a peak intensity of 30 counts at a 29.40° orientation, is in accordance with the published literature [7]. On the contrary, the collected precipitate from healing showed an intense peak of calcite. The obtained peaks in the XRD analysis of the healing product showed the most dominant peak of calcite (rhombohedral morphology) with a peak intensity of 174 at a 29.45° orientation. In the published literature, pure calcite XRD analysis resulted in a dominant peak at a 2-theta (2θ) diffraction angle of 29.30° [49]. The obtained peaks in our analysis resemble pure calcite, confirming the presence of calcite [2,7,10,49].

Further, the surface morphology and elemental composition of the healing precipitate of biodeposition treatment are displayed in Figure 17. Micrographs obtained from SEM analysis resulted in a dense calcite crystal with spherical morphology. The crystal morphology of calcite depends upon the food source and type of bacteria [3]. In this study, an irregularly stacked crystal of calcium carbonate with amorphous morphology was determined. This crystal of calcite perfectly resembles the calcite precipitate reported in the literature for *Bacillus* species with Calcium lactate [5,50]. After morphological confirmation, the elemental composition of these crystals was evaluated using spot-EDS (Figure 17). The elemental mapping obtained in the EDS study was primarily composed of high percentages of C, O_2,_ and Ca confirming the presence of calcite in healing material [2]. Small percentages of Zn, Si, and Mg were also detected, which could be due to the hydration product of cementitious constitutes in mortar. The high Ca content from the EDS of biodeposition healing products was also verified in the literature [5,10]. Thus, forensic evaluation using RAMAN, XRD, SEM, and EDS confirms the presence of calcite in the healing product of the repaired crack area using biodeposition repair with a Ca source.

Additionally, a thermal inspection of the healing product was conducted using TGA analysis and compared with the published literature. The monitored weight loss over time exactly resembles the published literature on calcite decomposition as shown via Figure 18. In our study, a sharp fall of 17.56% was observed between 600 and 750 °C. This sharp fall is due to the decomposition of calcite into CaO and CO_2_, which normally occurs between 600 and 850 °C for pure calcite. The thermal stability and resemblance of the healing product to calcite were verified by this decomposition trend. Additionally, the weight loss derivative was also plotted with temperature. The obtained trend shows a single peak of 0.1236%/°C at 680 °C, confirming the presence of calcite [5,26,51].

## 4. Conclusions

From the current investigation, the following conclusions can be drawn:As the number of cycles increases, mortar fractures treated with biodeposition with a Ca-lactate food source showed progressive crack healing. The majority of the samples had a healed area of more than 50% after seven cycles of repair. After 14 cycles, all the fractures were sealed with a calcite coating on the top surface. Samples treated with a bacterial solution and pure water, however, failed to exhibit any discernible crack healing.The biodeposition treatment significantly decreased the permeability of samples of cracked mortar. Before repair, the permeability of cracked mortar samples with average fracture widths of 0.29 to 1.08 mm ranged from 9.4603 × 10^−7^ to 1.8227 × 10^−4^ m/s. A reduction in permeability between 7.4396 × 10^−7^ and 9.3953 × 10^−6^ m/s was observed in mortar samples after 14 cycles of biodeposition repair. However, treatment with bacterial solution and distilled water alone did not significantly reduce sample permeability. This is due to the inefficiency of autogenous crack healing caused by cement hydration for larger crack widths.The ultra-sonic investigation of samples repaired by biodeposition with Ca-lactate ranged from 2116.67 to 3078.79 m/s before repair. After the application of 14 cycles of biodeposition repair, the elapsed ultra-sonic time reduced significantly due to crack closure. The sonic values of the repaired samples ranged from 2902.86 to 3907.69 m/s, showing a substantial increase in ultra-sonic value after repair, whereas samples after the application of seven cycles of bacterial solution repair and distilled water repair did not show any substantial change in ultra-sonic value.The recovered compressive strength of samples repaired by biodeposition with Ca-lactate ranged from 26.12 to 33.72 N/mm^2^. This shows recovered compressive strength of up to 79% of the uncracked mortar sample. However, the sample treated with a bacterial solution and distilled water resulted in only up to 55% and 56% compressive strength recovery, respectively.The forensic analysis of the healing product from biodeposition treatment with Ca-lactate confirmed the presence of calcites. The obtained RAMAN spectra were compared with the trends of pure calcite from the RRUFF database. The obtained vibrational modes, such as internal-mode *A_g_*, due to *v*_1_ stretching, had lower intensity *E_g_* due to in-plane bending mode *v*_4_; transitional and rotational lattice modes; lattice mode peaks at 1087 cm^−1^ and 710 cm^−1^, and below 300 cm^−1^, 282 cm^−1^, and 151 cm^−1^, respectively. These resultant modes and respective peaks resemble calcite from the published literature.The obtained XRD spectrum diffraction peaks of the healing material matched perfectly with the pure calcite diffraction peaks of calcite. Further, the XRD pattern “Joint Committee of Powder Diffraction Standards (JCPDS)” card number 5-586 resulted in Miller indices and planes belonging to calcite.The obtained SEM micrograph showed the dense spherical crystals of calcite, which was further confirmed by elemental mapping using spot-EDS.The thermogravimetric analysis (TGA) revealed that the thermal degradation of the healing product closely matched the documented calcite patterns.

## 5. Recommendations

The process of biological restoration by pairing a concrete-suitable bacterium strain with a Ca-based food supply is particularly useful since bio-mediated calcite precipitation is both pollution-free and natural. The biodeposition process is also simply a successful repair approach. This research should be expanded to include other microbial formulations and crack repairs for larger crack widths. Additionally, the effect of pH on calcification potential needs to be incorporated into future studies.

Further samples healed through bio-deposition treatment need to be tested against further durability and mechanical testing such as freeze and thaw, chloride penetration, fatigue testing, the stress–strain response, etc. Additionally, new microbial strains with an enhanced capacity to precipitate and to heal wider cracks should be tested. The forensic evaluation of calcite should be tested against FTIR and other microstructural studies.

## Figures and Tables

**Figure 1 materials-15-06616-f001:**
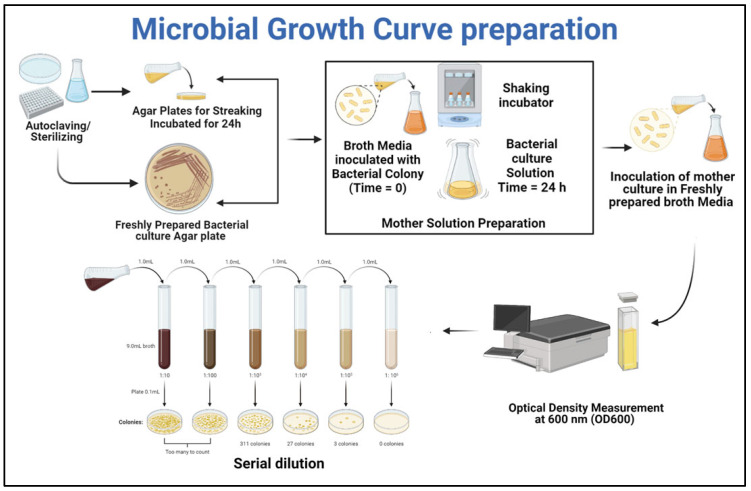
Microbial life-cycle assessment.

**Figure 2 materials-15-06616-f002:**
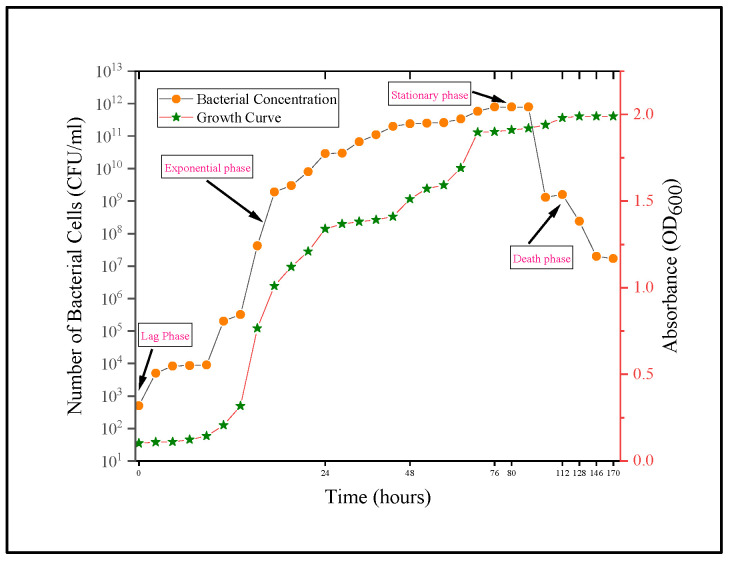
*Bacillus pumilus* growth curve.

**Figure 3 materials-15-06616-f003:**
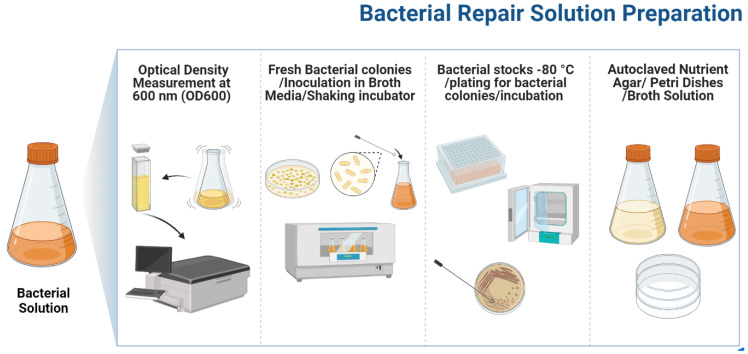
Biodeposition repair preparation.

**Figure 4 materials-15-06616-f004:**
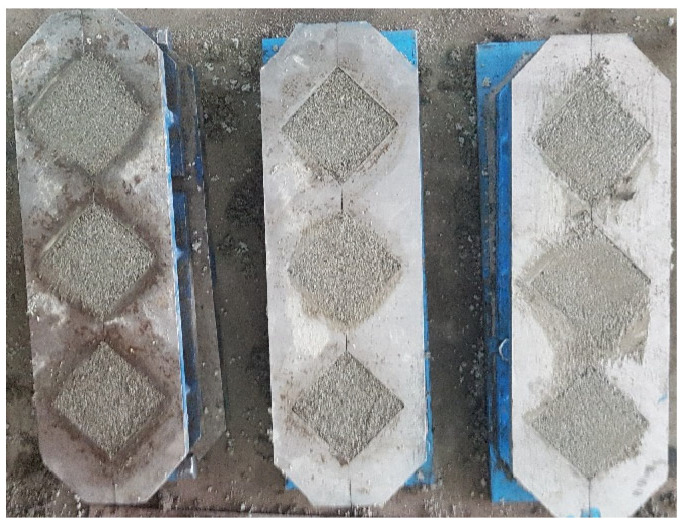
Preparation of Mortar samples.

**Figure 5 materials-15-06616-f005:**
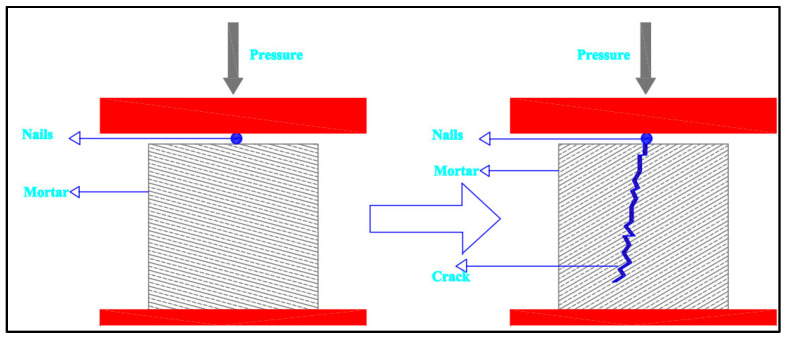
Mortar sample crack formation (schematic diagram).

**Figure 6 materials-15-06616-f006:**
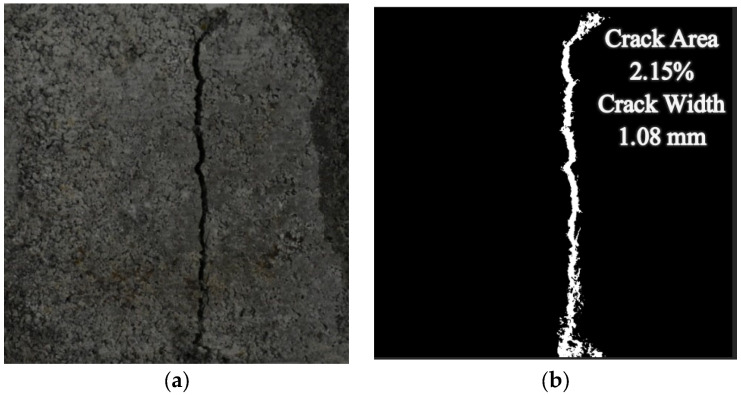
Cracked mortar sample. (**a**) RGB Picture, (**b**) binary Image.

**Figure 7 materials-15-06616-f007:**
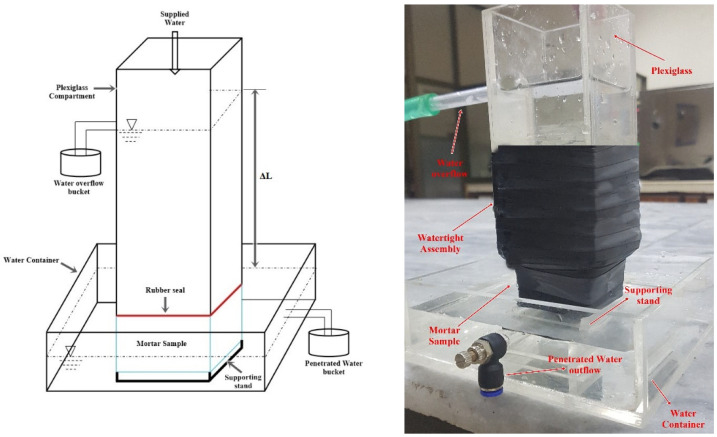
Permeability setup.

**Figure 8 materials-15-06616-f008:**
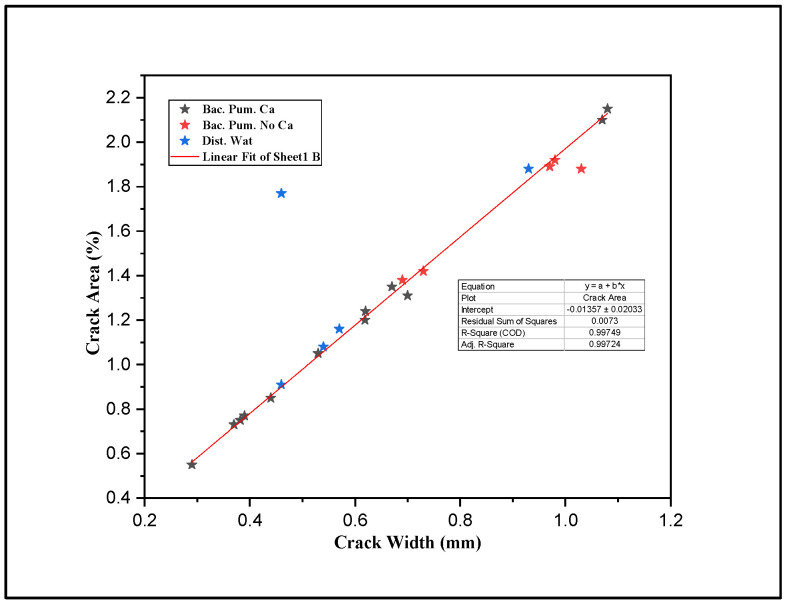
Image processing results: crack width vs. crack area.

**Figure 9 materials-15-06616-f009:**
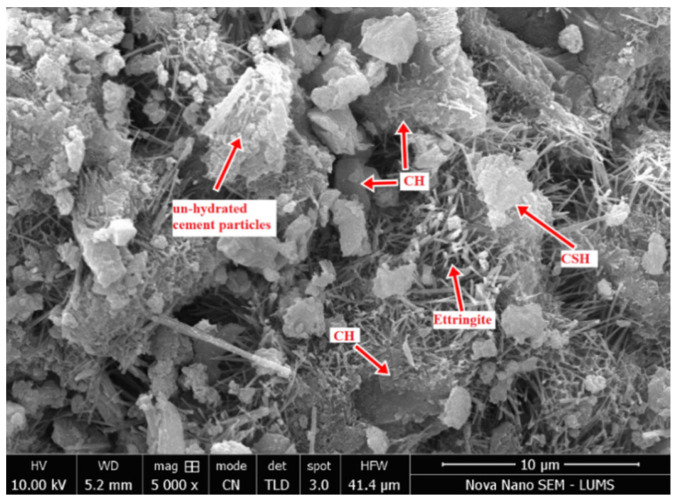
Micrograph of the pristine mortar sample.

**Figure 10 materials-15-06616-f010:**
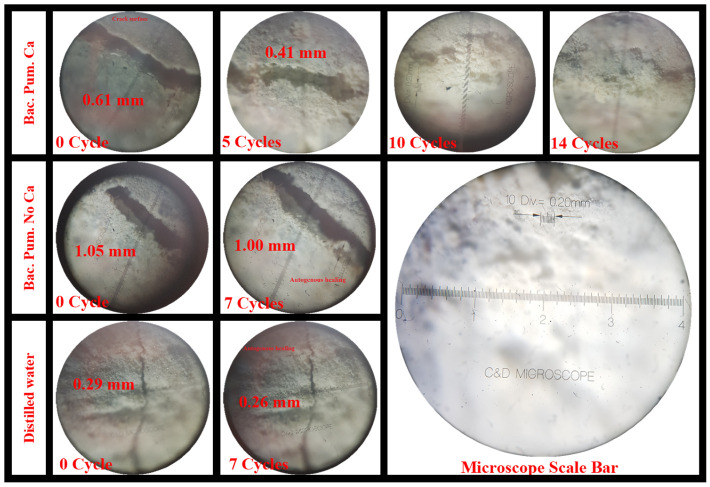
Crack healing under an optical microscope.

**Figure 11 materials-15-06616-f011:**
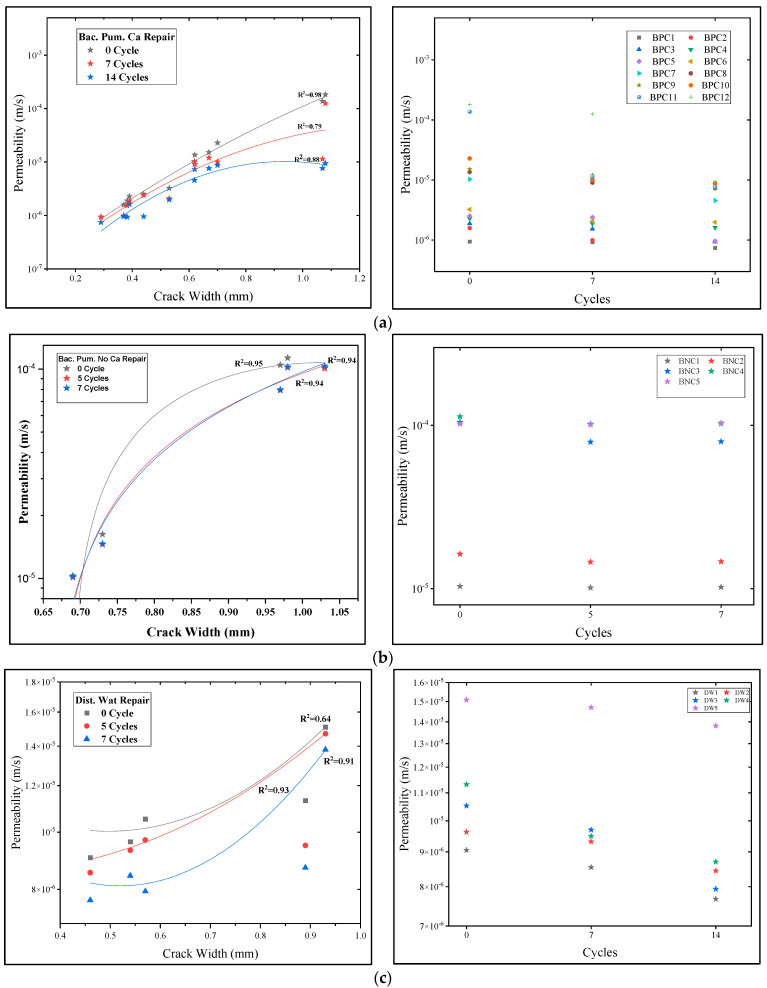
Relationship between repair cycles and permeability. (**a**) Effect of “Bac. Pum. Ca” repair, (**b**) effect of “Bac. Pum. No Ca” repair, (**c**) effect of “Dist. Wat” Repair.

**Figure 12 materials-15-06616-f012:**
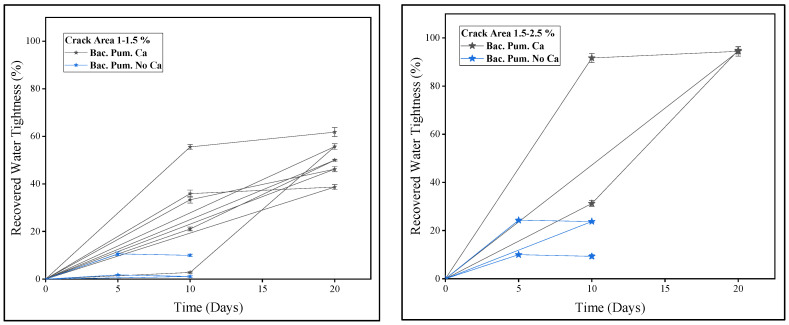
Recovered watertightness using biodeposition treatments.

**Figure 13 materials-15-06616-f013:**
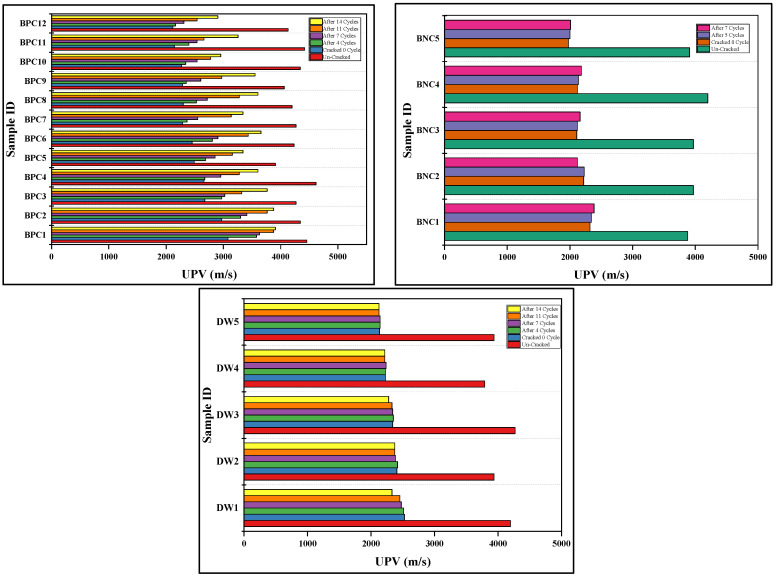
Relative change in ultra-sonic values after different repair treatments.

**Figure 14 materials-15-06616-f014:**
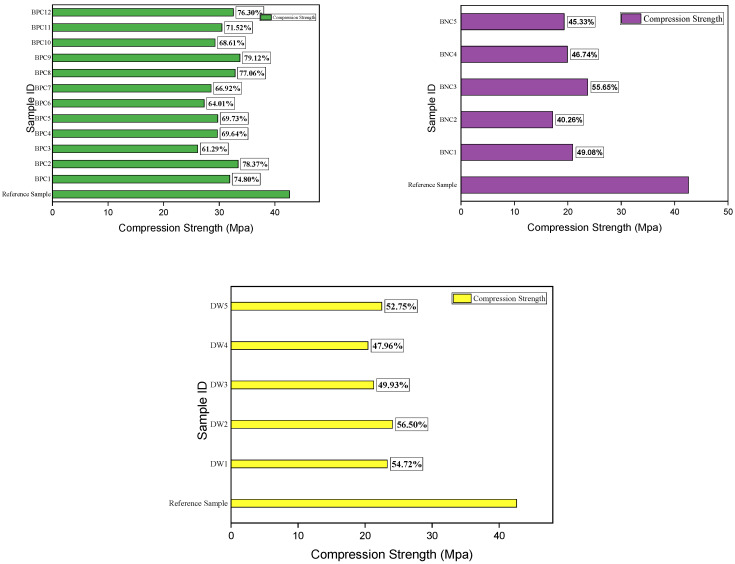
Recovered compressive strength after different repair treatments.

**Figure 15 materials-15-06616-f015:**
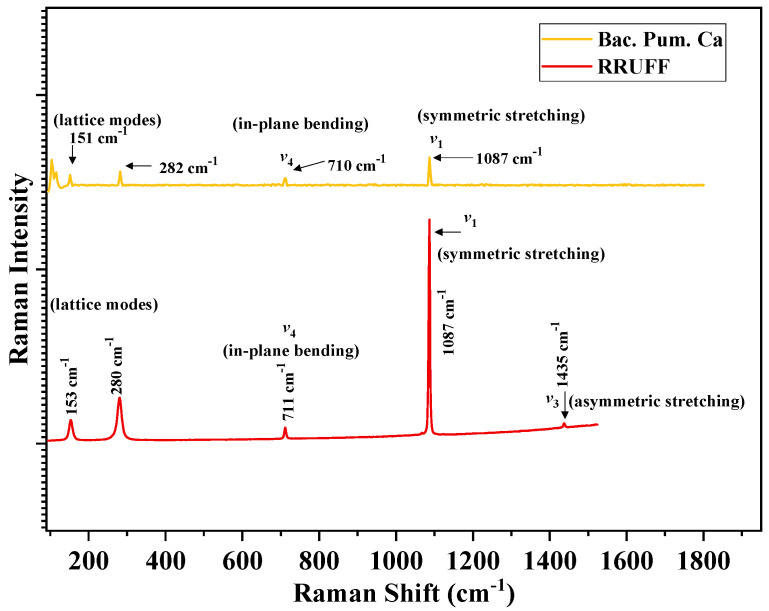
RAMAN spectra of healing precipitate from biodeposition treatment.

**Figure 16 materials-15-06616-f016:**
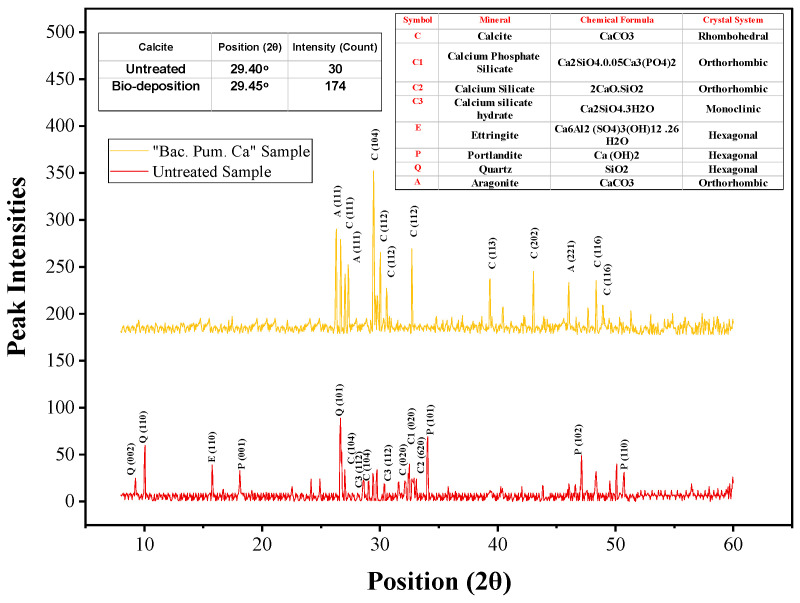
XRD of healing material.

**Figure 17 materials-15-06616-f017:**
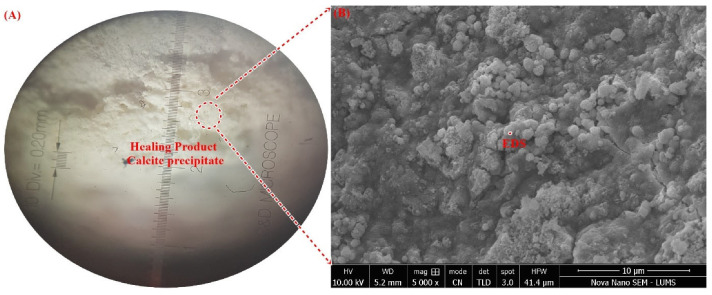
(**A**) Healed area under optical microscope, (**B**) SEM of healing product, (**C**) Spot-EDS of healing product.

**Figure 18 materials-15-06616-f018:**
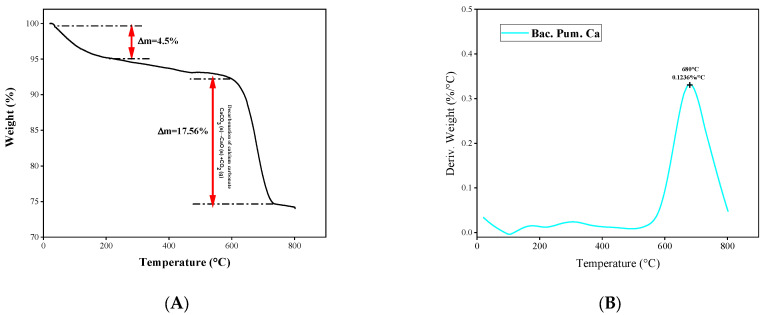
TGA plot of Healing product. (**A**) Weight loss versus Temperature, (**B**) derivative of weight loss versus temperature.

**Table 1 materials-15-06616-t001:** Percentage composition of ordinary Portland cement.

**SiO_2_**	**MgO**	**CaO**	**Fe_2_O_3_**	**Na_2_O**	**ZnO**	**K_2_O**	**SO_3_**	**Al_2_O_3_**	**P_2_O_5_**
18.3	1.5	61.29	3.20	0.76	2.1	0.97	2.51	7.31	0.089

**Table 2 materials-15-06616-t002:** Physical properties of OPC.

**Average Particle Size (µm)**	**Specific Gravity**	**Blain Fineness (cm^2^/gm)**	**Loss on Ignition**
9.5	3.14	1720	1.03

**Table 3 materials-15-06616-t003:** Physical properties of sand.

Fineness Modulus (FM)	Absorption (%)	Specific Gravity	Specific Gravity (OD)	Specific Gravity (SSD)
2.6	2.02	2.64	2.51	2.58

**Table 4 materials-15-06616-t004:** Crack repair compositions.

Sr.	Type	Treatment	Composition	Denotation	Sample IDs
					Realistic Crack
1	Bio-deposition ^a^	*Bacillus pumilus* ^b^	No calcium source	Bac. Pum. No Ca ^d^	BNC
			Calcium lactate ^c^	Bac. Pum. Ca	BPC
2	Soaking	Distilled water	--	Dist. Wat	DW
^a^	MICP				
^b^	The following nutrients were included in the broth solution for bacterial germination: 17 g/L pancreatic digest of casein, 3 g/L papaic digest of soya bean meal, 5 g/L sodium chloride, 2.5 g/L dipotassium hydrogen phosphate, and 2.5 g/L dextrose/glucose. The bacterium solution had an OD_600_ of 1.3 with 2.9 × 10^10^ cells/mL.				
^c^	Firstly, treatment with the bacterium solution of 100 mL for 4 h was carried out and we removed a sample from the bacterium solution. Secondly, we soaked the sample in calcium lactate solution for 24 h.				
^d^	The bacterial solution only.				

**Table 5 materials-15-06616-t005:** Mortar crack widths determined using optical microscopy and image processing.

Sample ID	Crack Width Determined Using Optical Microscope	Digital Image Processing (Average of Pixels)
	(mm)	(mm)
DW5	1.81	1.80
DW3	1.1	1.12
BPC7	1.12	1.11
BPC9	1.4	1.4
BNC3	1.31	1.29

## Data Availability

Data is contained within the article.

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
