# Peer review of "Bacterial Carbonate Precipitation Using Active Metabolic Pathway to Repair Mortar Cracks"

_materials, 2022, doi:10.3390/ma15196616_

Round 1

Reviewer 1 Report

What is Biodeposition treatment?

Introduction: to which category microbial healing belongs?

In chapter 2.1. , explained fig. 1,2 and 3.!

Tables 6 and 7 should go as soon as they are first mentioned in the paper.

The paper has a good scientific background but there is to many results and it is hard to follow. It is really confusing. Maybe to try to divide the work into two or three smaller and focus just on one thing and not to put everything in one work. Or take a smaller number of samples for testing. Readres lossing attention and interest.

Author Response

We thank the reviewer for their careful reading of the manuscript and their constructive remarks. We have taken the comments on board to improve and clarify the manuscript. Please find below a detailed point-by-point response to all comments. 

Reviewer 2 Report

This is intriguing research, as the authors have compiled a one-of-a-kind dataset using a Bacterial carbonate precipitation technique. Overall, the work is not nicely written and organised. However, I believe the study has major flaws in several data analyses and writing, and I believe this unique dataset has not been used to its best potential. I've made multiple comments on the text below since it's frequently ambiguous and long-winded. In a few cases, I also proposed citing more current and relevant material. Key critical points are-

The introduction part is too short. More information is needed in the Introduction part.

The introduction part looks like thesis data.

A lack of discussion about the results. Authors have not justified their results with previous research.

Improve the quality of figures.

The conclusion does not provide much details about future recommendations.

The paper is not written in a scientific way.

Lots of grammatical errors.

The figures are too small and not clear.

References are too old.

A revised manuscript in the current stage might not fit the journal. Given these shortcomings, the manuscript should be revised.

Author Response

(The authors gave the same response as above.)

Reviewer 3 Report

General Comments:

This paper deals with investigation of the robust microbial healing technique which is sustainable and durable in a high alkaline concrete environment without the use of an immobilizer. Especially it is considered that the originality of this paper is the selection of the Bacillus pumilus strain isolated from soil enough for its excellent performance in a highly alkaline environment and dense calcite production. Reviewer is not familiar with this field very much. However reviewer judges this paper is very interesting. It is considered that these results in this paper is very useful information for this area of this research field in the future. This paper provides valuable data for readers.

Then it is considered that there are several points that wants author to explain it in detail more

1)     Equation (7) at line 84 to 85:

Do the production of CO2 excessively reduce the pH of concrete cracked?

When there are reinforcing bars around this area, do the passivity film on these reinforcing bars destroy?

Does the carbonation of concrete not take place inside?

If explanation is possible, please describe the carbonation of CO2 exhausted from Bacillus pumilus.

2)     “2.3 Mortar Samples” of line 146 to 167:

There is no information of the consistency of mortar used in this study. How much the value of the slump flow was?

The relationship the position of the side induced the crack and the casting direction of test specimen  are not described in this part. Please explain it in detail more.

 3)     Line 211:

With the real concrete structure, it may not be 25 degrees Celsius. Particularly, will it be managed by this laboratory finding when it is lower than 25 degrees Celsius?

 4)     “4. Conclusions” of line 556 to 567:

It is considered that “Conclusionsare very redundant. Please revise it up briefly more.

Therefore this paper is required to be improved to be published in the Journal of “Materials”.

Author Response

(The authors gave the same response as above.)

Reviewer 4 Report

The article contains potentially interesting research findings and I therefore believe it is worth considering publication. However, it requires significant revisions and corrections.

1. the authors should clearly present the study plan, sample codes, etc. The methodology for testing the strength of specimens before and after repair is not clear.

2. the authors should refer to the problem of practical application of research results in the context of the research methodology adopted by them.

3. in my opinion, the form of presentation of research results should be improved. It concerns both the results themselves (huge number of pictures and drawings that are difficult to read) and the analysis of the research results. What is the variance of the obtained results?

4) Relationships on figures 14 and following need better analysis. 

5) very young concrete was used for repair. Please comment on how much the test results may apply to "old" concrete.

6) Overall the article needs to be rewritten to improve readability and shorten it.

Author Response

(The authors gave the same response as above.)

Round 2

Reviewer 2 Report

The authors have revised the manuscript as per the reviewer's comments.

Reviewer 4 Report

The authors have sufficiently considered my comments.